# ROS as Signaling Molecules to Initiate the Process of Plant Acclimatization to Abiotic Stress

**DOI:** 10.3390/ijms252111820

**Published:** 2024-11-04

**Authors:** Larisa Ivanovna Fedoreyeva

**Affiliations:** All-Russia Research Institute of Agricultural Biotechnology, Timiryazevskaya 42, 127550 Moscow, Russia; fedlara@inbox.ru

**Keywords:** reactive oxygen species, oxidative stress, abiotic stress, signaling wave, acclimatization

## Abstract

During their life cycle, plants constantly respond to environmental changes. Abiotic stressors affect the photosynthetic and respiratory processes of plants. Reactive oxygen species (ROS) are produced during aerobic metabolism and play an important role as regulatory mediators in signaling processes, activating the plant’s protective response to abiotic stress and restoring “oxidation-reduction homeostasis”. Cells develop normally if the rates of ROS production and the ability to neutralize them are balanced. To implement oxidation-reduction signaling, this balance must be disrupted either by an increase in ROS concentration or a decrease in the activity of one or more antioxidant systems. Under abiotic stress, plants accumulate excessive amounts of ROS, and if the ROS content exceeds the threshold amount dangerous for living organisms, it can lead to damage to all major cellular components. Adaptive resistance of plants to abiotic stressors depends on a set of mechanisms of adaptation to them. The accumulation of ROS in the cell depends on the type of abiotic stress, the strength of its impact on the plant, the duration of its impact, and the recovery period. The aim of this review is to provide a general understanding of the processes occurring during ROS homeostasis in plants, oxidation-reduction processes in cellular compartments in response to abiotic stress, and the participation of ROS in signaling processes activating adaptation processes to abiotic stress.

## 1. Introduction

As a result of environmental changes, plants require a system for the rapid transmission of information about damage or the occurrence of abiotic stress. A plant can sense changes in environmental conditions at the level of only a single organ before the entire plant. The signal generated by a sensory organ of the entire plant must quickly spread throughout the plant and trigger adaptation mechanisms [1,2]. Plants have several systems for rapid information transmission—Ca^2+^, electrical, hydraulic waves, and the so-called reactive oxygen species (ROS) wave. The ROS wave is a process of self-replication of ROS from cell to cell [3,4,5]. Plants are constantly exposed to stress, as a result, of environmental changes, so plants have developed a complex acclimation system. To activate this system, at least two types of signals are needed: the ROS wave, which spreads at a tremendous speed from the initial site of stressor impact throughout the plant, and a specific signal from the type of abiotic stressor. The ROS wave, caused by abiotic stress, triggers a cascade of intercellular communication interactions that spreads through various plant tissues and is capable of transmitting a signal over long distances [6]. However, it has been noted that various abiotic stresses can also lead to specific fluctuations in the ROS wave, causing certain specific rearrangements of transcriptomes and metabolomes [6]. The ROS wave is used for signaling and triggering plant adaptation mechanisms to abiotic stress. As a result of the ROS wave signaling, transcriptomic and metabolomic reprogramming of cells occurs. It is proposed that the ROS wave activates a general acclimation response, independent of specific abiotic stressors, which may induce broad transcriptome and metabolome reprogramming and render plants resilient to multiple different stresses.

The formation of free radicals in biological materials was discovered in the 1950s, and it was hypothesized that oxygen radicals were generated by enzymatic reactions in vivo. Oxygen radicals were thought to be cell damaging and to cause oxidative damage to DNA, proteins, lipids, and other cellular components [7]. However, more recent studies have revealed positive functions of free radicals, including their involvement in metabolic processes and signaling pathways [8].

ROS have been shown to be produced to initiate redox signals to regulate a wide range of cellular reactions essential for life [9]. The ROS-generated signaling wave activates the expression of genes responsible for maintaining cellular redox homeostasis. In addition, it has recently been shown that the process of cell proliferation and differentiation depends on the redox balance [10]. While cell death was previously thought to be a direct consequence of ROS-induced oxidative stress, it is now accepted that ROS is involved in the activation of both autophagy and programmed cell death (PCD) [11]. Thus, ROS act as important regulators of various physiological processes. Depending on the redox balance between ROS and detoxification systems in various subcellular organelles, ROS play a role either as oxidants or as redox signals [12]. The level of ROS is strictly controlled in space and time as a result of changes in the composition of ROS production and its detoxification [13]. In case of low ROS in the cell, they act as positive signaling molecules actively participating in many biologically important processes. With increasing ROS concentration, they can act as highly toxic molecules. The accumulation of ROS in the cell depends on many parameters. The complex process of plant adaptation to abiotic stress involves a whole range of oxidant–antioxidant interaction mechanisms capable of flexibly altering redox signals [14].

## 2. Reactive Oxygen Species (ROS) and Their Formation in Plant Cells

Oxygen is an essential element required for normal plant development. In the ground state, oxygen (triplet oxygen, ^3^O_2_) has two unpaired electrons, which are not reactive [15]. In the process of oxidation-reduction reactions, molecular oxygen is an electron acceptor, and it is able to attach from one to three electrons, which leads to the formation of reactive oxygen species (ROS). In biological systems, transition metal ions (Fe^2+^, Cu^+^) and semiquinones can act as electron donors, resulting in the activation of molecular oxygen [7]. The term ROS includes not only free radicals (superoxide radical (O^2•−^), hydroxyl radical (OH•), perhydroxy radical (HO^2•^), peroxyl (RO^2•^), carbonate (CO_3_^•−^), semiquinone (SQ^•−^), alkoxy radical (RO•), and peroxy radical (ROO•)) but also molecules such as hydrogen peroxide (H_2_O_2_), singlet oxygen (^1^O_2_), ozone (O_3_), organic hydroperoxide (ROOH), hypoiodous acid (HOI), hypobromous acid (HOBr), and hypochlorous acid (HOCl) (Figure 1) [16,17,18,19].

As a result of oxidation-reduction reactions occurring in various compartments of plant cells, ROS are formed due to an incomplete or partial reduction in oxygen molecules [20]. Several reactions associated with ROS formation are shown in Figure 2.

The mechanisms of ROS formation in biological systems are carried out in the course of non-enzymatic and enzymatic reactions. The mechanisms that trigger a certain type of reaction are regulated by the concentration of oxygen in the system. It is believed that at high concentrations of O_2_, ROS formation occurs via the non-enzymatic pathway, and at very low concentrations of O_2_ via the enzymatic pathway [21].

The most common ROS with high oxidizing capacity is the superoxide anion radical (O_2_^•−^). The superoxide anion is formed as a result of the monovalent reduction in the triplet state of molecular oxygen (Figure 2) [16]. Among the sources of ROS formed via the enzymatic pathway, mention should be made of xanthine oxidase (XO), the enzyme responsible for the initial activation of oxygen. XO can use xanthine, hypoxanthine, or acetaldehyde as electron donors [22]. The reactions of dismutation of O_2_ to H_2_O occur according to Figure 1.

The latter has been shown to accumulate under oxygen deficiency, especially during hypoxia [23].

The next important step in ROS conversion is the dismutation of the superoxide anion (O_2_^•–^) by superoxide dismutase (SOD) to form H_2_O_2_. The level of H_2_O_2_ is regulated by an enzymatic set of catalases (CAT) and peroxidases (PX), localized in almost all compartments of the plant cell. The main function of PX is the neutralization of H_2_O_2_; however, this family of enzymes can also catalyze the formation of O_2_^•−^ and H_2_O_2_ as a result of NADH oxidation. The oxidation of NADH with H_2_O_2_ produces the radical NAD•, which reduces O_2_ to O_2_^•−^, some of which is dismutated to H_2_O_2_ and O_2_ [24]. Thus, peroxidases and catalases play an important role in the fine regulation of ROS concentration in the cell through the activation and deactivation of H_2_O_2_ [16]. Alternatively, hydrogen peroxide can be converted to water by the enzymes catalase or glutathione peroxidase. In the glutathione peroxidase reaction, glutathione is oxidized to glutathione disulfide, which can be converted back to glutathione by glutathione reductase (GR) in a process that consumes NADPH, according to Figure 2 [25].

By scavenging O_2_^•−^, SODs reduce the risk of OH• formation via a metal-catalyzed Haber–Weiss-type reaction, as this reaction occurs 10,000 times faster than spontaneous dismutation [26]. The reaction results in the formation of H_2_O_2_, which is also an ROS with a dual role (toxicity and signaling). H_2_O_2_ can be effectively detoxified if the cellular antioxidant defense mechanism is sufficient. In some ways, this enzyme is unique in that its activity determines the concentrations of O_2_•− and H_2_O_2_, the two substrates of the Haber–Weiss reaction, and so it is likely to play a central role in the defense mechanism, according to Figure 3 and Figure 4 [27].

In biological tissues, superoxide can also be non-enzymatically converted to the non-radical species hydrogen peroxide and singlet oxygen (^1^O_2_) [28]. In the presence of reduced transition metals (e.g., iron or copper ions), hydrogen peroxide can be converted to the highly reactive hydroxyl radical (•OH).

Another aggressive oxidant is the hydroxyl radical (•OH), which is involved in oxidative damage to DNA bases [19]. •OH is produced by iron-catalyzed H_2_O_2_ and O_2_•− [29,30,31]. •OH can be formed by the ionization of H_2_O and also by photolytic decomposition of alkyl hydroperoxides. It is generally assumed that HO• in biological systems is formed by the redox cycle of the Fenton reaction (Schematic (4)), where free iron (Fe^2+^) reacts with hydrogen peroxide (H_2_O_2_) and the Haber–Weiss reaction, which results in the formation of Fe^2+^ when superoxide reacts with ferric iron (Fe^3+^) (Schematic (3). In addition to the iron redox cycle described above, a number of other transition metals, including Cu, Ni, Co, and V, may be responsible for the formation of HO• in living cells.

## 3. Oxidative Stress

Redox reactions are common in living things and are responsible for the majority of ROS generation [32]. Excessive ROS production, if not scavenged, can lead to so-called “oxidative stress”. Physiological levels and types of ROS in different compartments are highly dependent on the energetic load to which the cellular response is subjected. Even within a single cell, there are at least eight different organelle compartments (mitochondrial matrix, lysosomes, smooth ER/SR, rough ER, Golgi, peroxisomes, nucleus, cytosol), each with its own redox balance [33]. Therefore, the term oxidative stress should be used when the levels and types of oxidants in a cell or organelle, on average, significantly exceed the level associated with normal homeostatic function for each compartment. In plant cells, redox homeostasis is a normal state in which the antioxidant defense system aims to maintain a balance between ROS production and antioxidant activity [34]. Under normal conditions, ROS production and levels are safe for normal cellular function, including proliferation, differentiation, signaling, and intercellular communication [17,35,36]. The relationship between ROS and redox changes/regulation in cells is commonly referred to as redox biology and is thought to play a key role in ROS-mediated signaling and/or cellular metabolic regulation [6,37,38,39]. Because membranes can function as barriers to redox levels, each subcellular compartment can contain its own redox state that will correspond to its own steady-state ROS level, contributing to the formation of a cellular-specific ROS signature during abiotic stress [40].

Reactive oxygen species are a by-product of aerobic metabolism in various cellular organelles such as chloroplasts, mitochondria, peroxisomes, plasma membranes, and cell walls [19,41]. The specific generation of ROS in the cell is highly localized, and different pathways are extensively involved in this process [42,43]. To better understand the ROS scavenging tactics in different subcellular compartments, it is imperative to first study the subcellular compartment-specific ROS generation in cells.

ROS, as signaling molecules, play an important role in biological processes, the so-called “redox biology” [11,44,45,46]. For participation in the redox signaling process, the basal amount of ROS is crucial [11,46]. In case the levels of reduced glutathione are too high, so-called “reductive stress” can occur, which also has detrimental consequences for the cell [47,48]. However, under abiotic stress, when ROS levels in plants are above or below normal levels, a redox imbalance occurs that can cause impairment and/or even loss of certain cellular functions. Low ROS levels trigger signaling that alters normal plant metabolism, while excess ROS initiates oxidative damage to cells [11,31]. Therefore, the mechanisms that ensure a stable equilibrium between ROS production and its neutralization must function together and be harmonized [49,50].

Thus, redox homeostasis is a prerequisite for the normal functioning of cells and organisms [51]. It has been previously argued that the cell normally maintains cytosolic thiols in a highly reduced redox state, thus not supporting the existence of reductive stress [47]. Under stress conditions, ROS accumulation in plant cells disrupts the “redox state” of various proteins, enzymes, and receptors, altering or participating in various signal transduction pathways in response to an abiotic factor [46,49]. There is a growing understanding that the role of redox metabolism enzymes and metabolites goes far beyond a simple ROS scavenging function. First, as specialized processors of ROS signaling, these enzymes act as integral parts of a complex signaling system [52].

## 4. Regulation of ROS in Cellular Compartments

Reactive oxygen species are a by-product of aerobic metabolism in various cellular organelles such as chloroplasts, mitochondria, peroxisomes, plasma membranes, and cell walls (Figure 3) [19,41]. ROS generation is specific in different cellular compartments [42,53]. Abiotic stresses that limit CO_2_ availability through stomatal closure enhance the production of ROS such as O_2_^•−^ and ^1^O_2_ in chloroplasts, which in turn can initiate retrograde and anterograde signaling [54]. ROS signaling molecules are mainly produced in the apoplast by NADPH oxidases, some oxidases and peroxidases, and in other cellular compartments through various pathways [55,56,57,58]. The signaling as well as metabolic processes are represented as a continuous process of ROS production and scavenging, which occurs in all cellular compartments [49]. Each cellular compartment has its own ROS level, which determines its ROS signature and depends on the cell type and the type and strength of the stressor. With the help of a system of sensors that decipher the cellular signature, the mechanisms of adaptation to various abiotic stresses are activated.

### 4.1. Regulation of ROS in Chloroplasts

In plants, photosynthesis takes place in chloroplasts. Chloroplasts produce ROS via both photosystems, PSII and PSI [19,59]. Excessive light absorption in photosystem PSII results in the formation of a triplet state of chlorophyll (3Chl), which inhibits photosynthesis and promotes excessive ROS formation [60,61]. PSI and PSII in chloroplasts are the main systems producing ^1^O_2_ and O_2_^•−^ [7]. Chloroplasts produce not only ROS but also other stress-related signaling molecules, such as reactive carbonyl, nitrogen, and sulfur compounds, as well as precursors of stress hormones. Stress-related signaling pathways have allowed chloroplasts to be considered as specific markers that transduce stressor exposure into various signaling pathways. [7]. However, the amount of O_2_^•−^ generated in PSI by the Mehler reaction is converted to H_2_O_2_ by SOD. ROS is generated in chloroplasts in the reaction centers via PSI and II by trapping excess photons in PSII and scavenging electrons by molecular oxygen via PS [54,62,63]. Excess energy in PSII leads to the formation of the triplet state of ^3^Chl^.^ in the antenna complex of PSII. In PSII, ^1^O_2_ is formed due to energy transfer, as well as the formation of O_2_^•−^, H_2_O_2_, and OH^•^ [64,65,66,67,68]. In the absence of stress, electrons leak from the excited PS, reducing NADP to NADPH, which then enters the Calvin cycle where CO_2_ is reduced as an electron acceptor. Conversely, the electron transport chain (ETC) is overloaded, resulting in electron leakage from ferredoxin to O_2_, forming O_2_^•−^ [69]. Under high light intensity with low CO_2_ demand, direct electron transfer to molecular oxygen occurs due to stomatal closure via the Mehler reaction in PSI [70]. In PSI, the conversion of O_2_^•−^ to H_2_O_2_ depends on inefficient photochemical and non-photochemical quenching [70]. These superoxide radicals are converted to H_2_O_2_, which is further converted to water [71].

The chain of O_2_^•−^ conversion to O_2_ and H_2_O_2_ via SOD catalysis and H_2_O_2_ detoxification via the ascorbate–glutathione cycle by ascorbate peroxidase (APX) associated with the thylakoid membrane [62], called the water–water cycle, is critical to maintaining the redox balance in the electron transport chain for maximum photosynthetic efficiency [72]. Under various stress conditions, inefficient photochemical quenching occurs, resulting in excess energy in the PSII photosystem, which promotes the formation of ^1^O_2_. Excess electrons from the electron acceptor PSII result in the formation of O_2_^•−^, which is converted to H_2_O_2_. H_2_O_2_ is then converted to OH• by non-heme iron via the Fenton reaction in the presence of reduced metal ions such as Cu^+^ or Fe^2+^, which are formed by the reduction of Cu^2+^ and Fe^3+^ (Schematic (4)). Incomplete oxidation of water in photosystem II also results in the formation of H_2_O_2_, which is then reduced to OH• [67]. Although chloroplasts have an extensive antioxidant and scavenging system to maintain ROS homeostasis, severe or prolonged exposure to stressors can deplete and inactivate the function of these systems.

Consequently, unfavorable conditions lead to overproduction of ROS and increased damage to chloroplasts due to the reactivity of ROS towards macromolecules in their vicinity.

### 4.2. Regulation of ROS in Mitochondria

Photorespiration and oxidative phosphorylation also lead to ROS generation in mitochondria and peroxisomes. In the non-green parts of plants, mitochondria are the main site of ROS generation [73]. ROS produced in mitochondria reduce both mitochondrial energy transport and other subcellular functions. Respiratory complexes I and III are the main sources of mitochondrial ROS, especially O_2_^•−^. Next, O_2_^•−^ formed in both complexes is converted into H_2_O_2_ as a result of catalysis by MnSOD and Cu/ZnSOD dismutases [74,75].

ROS production in mitochondria is lower than in chloroplasts. The mitochondrial respiratory chain was originally described as flavin and cytochrome c-containing proteins in the inner matrix of mitochondria [76,77]. This model proposed that the four major complexes, i.e., NADH-coenzyme Q reductase (complex I), succinate coenzyme Q reductase or succinate dehydrogenase (complex II or SDH), ubiquinol cytochrome c reductase (complex III), and cytochrome c oxidase (complex IV) of the respiratory chain, are randomly distributed in the matrix and linked by the redox active enzymes coenzyme Q and cytochrome c [78]. This model was refined with complexes I, III, and IV, forming supercomplexes that allow efficient electron transfer with minimal O_2_^•−^ formation. However, ROS from complexes I, II, and III are not only involved in the random release of electrons from the ETC and their transfer to molecular oxygen but are now considered important mediators in physiological cell signaling. ROS production must be strictly controlled to avoid its overproduction, which can cause damage to mitochondrial and extramitochondrial macromolecules and cause cell death [79].

Under abiotic stresses, such as high temperatures and/or drought, respiration rate is high and transpiration and photosynthesis are low, increasing the need for mitochondrial ATP to compensate for ATP production by chloroplasts [52]. In mitochondria, O_2_ is reduced to O_2_^•−^ by NADPH dehydrogenase [complex I] [78]. Mitochondria have developed their own system for detoxifying excess ROS products, consisting of alternative oxidase (AOX) and MnSOD [73,79]. The presence of Mn as a heme in the superoxide dismutase enzyme indicates its localization in the mitochondria. The largest amount of ROS in mitochondria exists in the form of O_2_^•−^ molecules, which were converted by MnSOD to H_2_O_2_ and then by ascorbate peroxidase (APX) to O_2_ [80]. AOX is also involved in the detoxification of ROS in mitochondria, as shown by Giraud et al. [81] on *Arabidopsis* AOX mutants that exhibited high sensitivity to drought and light stress. As in chloroplasts, mitochondria produce ROS in the unstressed state at basal levels, and any stress has been shown to also reduce ROS levels by upregulating ATP synthesis [73,74,82,83].

### 4.3. Regulation of ROS in Peroxisomes

Peroxisomes contain a complex of oxidases that catalyze reactions that form H_2_O_2_ and O_2_^•−^. Glycolate oxidase (GOX) is capable of regulating gas exchange through the closing and opening of stomata [84]. Closing of stomata leads to a decrease in the flow of CO_2_, disruption of the photosynthesis process, and an increase in the production of H_2_O_2_ [85]. As in mitochondria and chloroplasts, O_2_^•−^ is produced in the peroxisome during normal metabolism, and its level is strictly controlled [86]. Xanthine oxidase (XOD) is responsible for the generation of O_2_^•−^ in the peroxisomal matrix, converting xanthine and hypoxanthine into uric acid and O_2_^•−^, using NADH and cytochrome b as electron acceptors [87]. Not only SOD metalloenzymes but also urate oxidase are responsible for the dismutation of O_2_^•−^ to H_2_O_2_ in peroxisomes [88,89]. Metabolic reactions that result in the formation of H_2_O_2_ in the peroxisome include β-oxidation, radical disproportionation, and the flavin oxidase pathway [43,90,91]. Peroxisomes have a well-developed system for detoxifying excess H_2_O_2_, including CATs, APX, and the ascorbic acid (AsA)-GSH complex [49,92,93]. It has been found that a decrease in AsA-GSH levels can lead to lipid peroxidation in peroxisomes. In addition, POX, a polyamine catabolism enzyme, is localized in peroxisomes and regulates genes that promote the formation and removal of ROS [29,42,94].

### 4.4. Regulation of ROS in the Apoplast

In the Arabidopsis apoplast, NADPH oxidases, which are activated in the stomatal guard cells, are involved in the generation of ROS production [95,96]. The activity of NADPH oxidases has been shown to be regulated by two genes, AtRbohD and AtRbohF [97,98]. In addition to NADPH oxidases, peroxidases, cell wall-associated oxidases, polyamine oxidases, and oxalate oxidases, which release H_2_O_2_ and CO_2_ from oxalic acid, play a role in the formation of H_2_O_2_ in the apoplast [80,99,100].

These oxidases induce oxidative deamination of polyamines with the help of cofactors. As observed by Heyno et al., the generation of hydroxyl ions in the apoplastic region of the cell completely or partially promotes the formation of cell wall-associated peroxidases [101]. Increased H_2_O_2_ production leads to increased levels of polyamines and Ca^2+^. This leads to increased production of H_2_O_2_, which activates the antioxidant mechanism. This activates the synthesis of polyamines and secondary messengers, such as Ca^2+^. In response to abiotic stress, polyamines activate signaling pathways triggered by ABA [94].

Various enzymes promote ROS formation in the apoplast, among which the most important are quinine reductase, NADPH oxidase, SOD, and PX [18,19]. Some apoplastic enzymes can also lead to ROS production under normal and stress conditions. Other oxidases responsible for the transfer of two electrons to oxygen (amino acid oxidases and glucose oxidase) can promote the accumulation of H_2_O_2_. Also, extracellular germline-like oxalate oxidase catalyzes the formation of H_2_O_2_ and CO_2_ from oxalate in the presence of oxygen [22]. Amine oxidases catalyze the oxidation of biogenic amines to the corresponding aldehyde with the release of NH_3_ and H_2_O_2_. Data on the accumulation of polyamine (putrescine) during oxygen starvation in rice and wheat shoots [102] and the predominant localization of amine oxidase in the apoplast suggest the participation of amine oxidase in the production of H_2_O_2_ during oxygen starvation.

### 4.5. Regulation of ROS in Cell Walls and Plasma Membranes

Plant cell walls accumulate oxidative radicals OH^•^, O_2_^•−^, H_2_O_2_, and ^1^O_2_ under stress. Peroxidases, lipoxygenases, oxidases, and polyamines localized in the cell walls are responsible for ROS formation. The formed oxygen radicals participate in the peroxidation of polyunsaturated fatty acids of lipids localized in the plant cell wall [103]. Generation of ROS by cell wall-associated peroxidases triggers a cascade of reactions in response to stress [43,104]. The role of polyamines in the detoxification of ROS formed during the plant response to stress has been established [105]. Therefore, it was suggested that exogenous polyamines are used to activate antioxidant processes in situ in plants in case of exposure to stressors [42,82,106]. NADPH oxidases in plasma membranes also play an important role in plants by converting O_2_ to O_2_^•−^ in response to stress factors [82,104]. It has been found that in some cases, other oxidases, such as quinone reductase, act in conjunction with NADPH oxidase to facilitate the conversion of O_2_ to O_2_^•−^ [82,104,106].

In the endoplasmic reticulum (ER), both O_2_^•−^ and H_2_O_2_ are generated by GOX and urate oxidase [107]. In addition, small amounts of O_2_^•−^ can also be produced by oxidative processes involving cytochrome P450 and cytochrome P540 reductase in the presence of reduced NADPH [43]. In the cytosol, the redox balance is maintained mainly by cytoplasmic NADPH, resulting in lower levels of ROS production than in other cellular compartments. Since the cytosol transmits ROS signals from various cellular organelles, modulating gene expression in the cell nucleus, the cytosol plays a key role in the redox signaling process in plant cells [108].

## 5. ROS Signaling Mechanisms

Abiotic stresses lead to increased production and accumulation of ROS, including H_2_O_2_. ROS signaling is mediated by oxidative reactions of proteins in response to highly reactive ROS. Oxidation of a wide range of proteins mediates signaling in response to a variety of stressors, which distinguishes them from the actions of signaling molecules and phytohormones [46]. H_2_O_2_ oxidizes thiol groups (–SH) of cysteine (Cys) residues in proteins. The formed –SOH oxidative modifications of proteins can alter protein functions by activating or inhibiting enzyme activity, thereby changing their subcellular localization and interactions with other proteins. Oxidized proteins can be reduced by catalysis of GRX, PrxR, and thioredoxin (TRX). However, excess ROS can lead to further oxidation of proteins with –SOH groups to sulfinic or sulfonic acids, which can lead to protein dysfunction and degradation according to Figure 5.

Thiol redox regulation plays an important role in plant responses to environmental stresses. Recent studies have shed light on the role of H_2_O_2_-oxidized proteins as sensors or receptors in plant growth and stress responses.

Most studies have been conducted on H_2_O_2_-oxidized proteins as sensors or receptors in plant growth and stress responses. For example, oxidation of some specific chloroplast proteins during light–dark cycles [109,110,111] formed plasma membrane-localized leucine-rich-repeat kinase HYDROGEN-PEROXIDE-INDUCED Ca^2+^ INCREASES 1 (HPCA1), which upon oxidation of H_2_O_2_ formed two intermolecular disulfide bridges to facilitate the movement of Ca^2+^ ions [112]. HPCA1 also acts as a central ROS receptor required for cell-to-cell ROS signaling, systemic signaling in response to various abiotic stresses, stress responses at local and systemic scales, and plant acclimation to stress [113].

## 6. ROS Under Abiotic Stress

Abiotic stresses are closely related to climate change and hinder plant growth and development; hence, they also negatively affect crop yield and quality [18,114]. Environmental impacts on plants can be both long-term and short-term and vary in strength. Depending on the set of mechanisms launched by plants in response to stress, which can vary significantly for each plant species and even for different varieties of the same species, their tolerance to stress and rapid acclimatization can be determined [115]. During abiotic stress, ROS production is impaired, resulting in disruption of ROS metabolic and signaling pathways. Metabolic ROS can directly alter the redox status of enzymes involved in the metabolism process, which may lead to changes in the metabolic rate. Alteration of metabolic reactions in the cell may negatively affect defense mechanisms to abiotic stresses [116].

The steady-state ROS level and redox state of the plant cell differ depending on the type of abiotic stress the plant faces. It follows that different environmental conditions lead to the production of a specific set of subcellular ROS and redox signatures, which in turn leads to the activation of an acclimation response adapted to them.

Gas exchange and transpiration in plants are carried out by stomata, specialized cells of the epidermis. In recent years, several studies have shown that plant RBOHs (respiratory burst oxidase homologs) are involved in a variety of different signaling pathways, including root hair growth, stomatal closure, pollen–stigma interactions, plant defense, and adaptation to various abiotic stresses. Integral RBOH proteins are localized in the plasma membrane. They contain hemes involved in electron transfer. [117]. At the N-terminus, there are two calcium-binding EF motifs and two phosphorylation sites, which are involved in the regulation of the enzyme activity [118]. RBOHs have a cytosolic N-terminal extension consisting of two Ca^2+^-binding EF hand motifs and target phosphorylation sites, which are important for their activity [119,120]. Upon activation, superoxide O_2_^•−^ is produced in the apoplast via the function of RBOH proteins and dismutated into H_2_O_2_ spontaneously or catalytically by superoxide dismutase (SOD) [121,122]. Membrane-permeable H_2_O_2_ may then play a key role as a signaling molecule that regulates cellular metabolism involved in growth, development, and response to environmental stimuli [123]. Mutations in the Ca^2+^-binding EF motifs of RBOH were found to result in impaired ROS production [124]. Furthermore, ROS production depends on RBOH phosphorylation and other modifications and thus shapes the ROS signature in the apoplast [125,126]. Various receptor-like kinases are actively involved in signaling from extracellular ROS.

RBOHs function in response to various stressors, but each has different physiological consequences. The plasma membrane-localized protein kinase GUARD CELL HYDROGEN PEROXIDE-RESISTANT 1 (GHR1) senses RBOH-derived ROS signaling and regulates stomatal movement [127].

Excess ROS and ROS modification lead to changes in the redox balance in the cell, which may alter the function of key regulatory proteins regulating transcription and/or translation [8,37]. In contrast, signaling ROS directly alter the redox state of regulatory proteins and alter transcription and translation, resulting in activation of acclimation responses to abiotic stresses, counteracting their negative impact on metabolism and reducing the level of metabolic ROS. Signaling ROS are produced in the apoplast [128]. These signaling ROS are then translocated into the cytoplasm via regulated aquaporins [129] where they alter the redox status of key regulatory proteins such as TFs that influence gene expression (Figure 4) [39,84,130].

As signaling molecules, ROS are distributed in all metabolically active plant tissues and are controlled by the ROS gene network [5,131,132]. ROS, together with Ca^2+^ ions, participate in long-range systemic signaling, participating in the activation of acclimation to abiotic stresses [52,107]. Plant hormones such as abscisic acid (ABA) and jasmonic acid (JA) trigger ROS production, initiating a systemic signal (ROS wave) [133]. Hormonally activated ROS move in a loop and activate acclimation mechanisms [134,135]. This mechanism is realized in a feedback loop that activates ROS and calcium, thus causing a response of the whole plant to acclimate to abiotic stresses.

Under stress conditions, compartments such as the chloroplast, peroxisomes, and mitochondria have been shown to exhibit expansion of membrane structures that contact the nuclear envelope and can directly alter the ROS state of nuclei [37]. Metabolic and signaling ROS can be produced in different subcellular compartments (e.g., metabolic ROS in the chloroplast and signaling ROS in the apoplast). However, they can influence each other’s levels and even shuttle between compartments [129]. However, more and more studies have shown that ROS play a dual role in plants [6,11]. An important positive role of ROS is to scavenge excess electrons in the chloroplast from the photosynthetic apparatus, thereby preventing antenna overload and subsequent damage. A similar scavenging function is also performed by ROS in the mitochondria. In plant cells, there are several levels of ROS detoxification pathways and mechanisms, which allow ROS to redirect electron flow and prevent overload of various cellular systems during stress [54]. ROS, as well as their ability to convert into other forms of ROS, may mediate the regulation of metabolic fluxes during stress, preventing damage or excessive accumulation of toxic products. The most beneficial role of ROS during abiotic stress is shown in signaling processes that activate acclimation processes (Figure 4) [8,39,56,58,136]. Plants with impaired ROS production or ROS scavenging were found to be more sensitive to abiotic stresses and also failed to mediate systemic signaling during abiotic stresses [55,137,138]. However, these useful functions of ROS can be demonstrated while the cell detoxifies excess ROS. In case of disruption of the detoxification system, as well as in case of significant and prolonged abiotic stress, excess ROS is extremely toxic for the cell, leading to the destruction of biomolecules and cell death. Thus, ROS molecules are important molecules regulating metabolism and triggering signals for acclimation processes in response to abiotic stress.

### Two Phases of ROS Formation Under Abiotic Stress

Plants have complex acclimation and defense mechanisms that can be activated in stressed tissues. However, cell-to-cell communication also activates defense or acclimation mechanisms in unstressed tissues, which is called systemic acquired resistance (SAR) or systemic acquired acclimation (SAA) [139,140].

How are the signals generated in local and systemic tissues related? The ROS wave may play a key role in spreading signals from local to systemic tissues. This leads to the accumulation of ROS in the apoplast (RBOHD produces ROS in the apoplast [55]), which is sensed by neighboring cells and causes them to increase ROS production via their own RBOHD proteins. This, in turn, causes neighboring cells to do the same, leading to a self-sustaining process of increased ROS production that can spread throughout the plant [4,133,141,142], Thus, an initial abiotic stress-induced ROS burst in a local group of plant cells triggers a cascade of intercellular communication events that spreads throughout different plant tissues and carries a systemic signal over long distances [142]. The ROS wave functions as a general priming signal in plants, alerting systemic tissues to the occurrence of a localized abiotic stress stimulus. Furthermore, we found that plant SAA to abiotic stress is mediated by the temporal–spatial interaction of the ROS wave with stress-specific hormones or amino acid signals activated in systemic tissues. ROS production has been demonstrated to consist of a primary phase that occurs within minutes and a secondary phase that occurs within hours/days [6,143,144]. This biphasic production of ROS accompanies several different signaling events in many biological systems [6,143,144]. Such two-phase formation of ROS may have practical significance for agricultural plants; by influencing some organs with a stressor, it is possible to stimulate the adaptation of the whole plant to the stressor.

## 7. Conclusions

Abiotic stresses, both climate change and intense, impede plant growth and negatively affect crop yield and quality. To cope with the constant environmental impact, plants have developed a complex acclimation system. The acclimation system is triggered by a cascade of signaling reactions. ROS formation in plants is a natural metabolic process. ROS play both positive and negative roles. At low concentrations, ROS are signaling molecules that activate proliferation–differentiation processes. During abiotic stress, ROS production is impaired, which leads to disruption of ROS metabolic and signaling pathways. The steady-state level of ROS and the redox state of the plant cell differ depending on the type of abiotic stress the plant faces. It follows that different environmental conditions lead to the formation of a specific set of subcellular ROS and redox signatures, which in turn leads to the activation of an acclimation response adapted to them. Thus, ROS is an essential molecule, signaling about abiotic stress and triggering the acclimation process. At the site of exposure to the stressor, ROS are produced, which trigger a wave of ROS that spreads at a tremendous speed throughout the plant. The ROS wave is used for signaling and triggering the mechanisms of plant adaptation to abiotic stress. As a result of ROS wave signaling, transcriptomic and metabolomic reprogramming of cells occurs.

However, the processes of ROS interaction with other molecules, signal transduction mechanisms, detection of signaling cascades, participation of ROS in metabolic processes, and regulation of genes involved in the processes of plant acclimation to various types of abiotic stress are complex and require further research.

## Data Availability

No new data were created or analyzed in this study.

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
