# Peer review of "ROS as Signaling Molecules to Initiate the Process of Plant Acclimatization to Abiotic Stress"

_ijms, 2024, doi:10.3390/ijms252111820_

Round 1
Reviewer 1 Report
Comments and Suggestions for Authors
The manuscript descripted the function and process of ROS reaction in plant. The manuscript is well organized.
Major revision:
There has less descript about abiotic stress to ROS of plant, especially the signal transduction pathway. How these six abiotic stresses in the manuscript induce ROS and the mechanism are not mentioned.
Minor revision:
In fig.1, SQ not SO
Author Response
There has less descript about abiotic stress to ROS of plant, especially the signal transduction pathway. How these six abiotic stresses in the manuscript induce ROS and the mechanism are not mentioned.
Response: Thank you for your valuable comments. I tried to add some chapters, especially about signaling pathways and some signaling mechanisms. In the review, I planned to touch only on the general signaling mechanisms under abiotic stresses. Despite a significant amount of research in the field of ROS, such questions as how plants determine what stress affects them and what ROS wave to launch to initiate the acclimation process are still unclear, so I did not discuss each abiotic stress separately.
In fig.1, SQ not SO
Reply: Thanks for the note. The error has been corrected

Reviewer 2 Report
Comments and Suggestions for Authors
The review article titled “ROS as Signaling Molecules to Initiate the Process of Plant Acclimatization to Abiotic Stress” discusses the crucial role of reactive oxygen species (ROS) in plant acclimatization to abiotic stress. In this study, ROS and cellular oxidative-reductive homeostasis are explored in a complex way. It demonstrates how ROS can be harmful at high concentrations, but can also serve as vital signaling molecules when controlled. Review discusses the formation of ROS and the role they play in orchestrating plant responses to environmental stress. Revisions can improve this manuscript's contribution to the literature
Comments to authors:
1. The review discusses ROS well despite their dual roles. Although some sections could be clarified more concisely, especially those dealing with ROS formation in cellular compartments. The readability of complex biochemical processes would be enhanced if they were simplified or divided into shorter paragraphs.
2. Several figures illustrate the formation and reactions of ROS in the article. I would appreciate if these diagrams were clearer. Provide comprehensive captions for all figures and refer to them consistently throughout the narrative.
3. Use consistent technical terms throughout the article. The term "ROS wave" is mentioned several times without explanation. Readers unfamiliar with the specific jargon might find it helpful to have a glossary.
4. Although the article discusses many ROS-related processes, the transition between sections could be improved. A sentence describing ROS formation in a single compartment of the cell may also help guide the reader through the narrative. The roles they play in signaling could also be described by these sentences. Furthermore, some sections, like those discussing ROS regulation in mitochondria and chloroplasts, seem repetitive.
5. The conclusion should emphasize specific gaps in knowledge regarding ROS signaling and plant stress responses. As opposed to briefly mentioning the need for further research.
6. There are many relevant studies cited in the review, but some are out of date. Incorporating more recent research, particularly from the past five years, would improve the review's accuracy.
7. Make your paper more appealing to a wider audience by emphasizing the practical implications of the findings. Learn how ROS mediated acclimatization can be used to develop crops that are stress-resistant or improve farming practices under harsh conditions.
8. Due to the length and complexity of the sentences in this article, readers may have difficulty following the main ideas. Condensing and simplifying sentences will enhance reading. It is sometimes difficult to agree on the subject-verb agreement as well as verb tense consistency. Consistently use past and present tenses if the article discusses past research findings rather than general principles.
Comments on the Quality of English Language- English Language Improvement Suggestions:
- Sentence Structure and Flow: Many sentences in the article are quite long and complex, making it difficult for readers to follow the main ideas. Shortening sentences and breaking them into simpler, more concise structures will enhance readability. For example, compound sentences could be split into two or more sentences for clarity.
- Grammar and Syntax: There are occasional issues with subject-verb agreement and verb tense consistency. Review the article for consistent use of past and present tenses, especially when discussing past research findings versus general principles. For instance, the phrase "ROS are produced" should be consistent with either "ROS is" or "ROS are," depending on whether ROS is treated as singular or plural.
- Punctuation: The article would benefit from improved punctuation, especially in the use of commas to separate clauses. In some instances, missing commas make it hard to distinguish between dependent and independent clauses, affecting sentence clarity. Proper placement of commas and periods will improve the natural flow of the text.
Author Response
Thank you very much for your kind review of the manuscript and for your valuable recommendations.
Each author has his own preferences, choosing the most important points in his opinion. As you understood, in my manuscript I made a special emphasis on oxidation-reduction reactions, the formation of ROS in cellular compartments. In different compartments there are both common mechanisms of ROS formation and their own characteristics.
The review discusses ROS well despite their dual roles. Although some sections could be clarified more concisely, especially those dealing with ROS formation in cellular compartments. The readability of complex biochemical processes would be enhanced if they were simplified or divided into shorter paragraphs.
Response: I completely agree with you that the material is more accessible when it is divided into small chapters. In my manuscript, I tried to divide the material not only into small chapters, but also to make sure that these chapters were united in meaning
Several figures illustrate the formation and reactions of ROS in the article. I would appreciate if these diagrams were clearer. Provide comprehensive captions for all figures and refer to them consistently throughout the narrative.
Response: Thank you for your comments and suggestions. I tried to take advantage of them.
Use consistent technical terms throughout the article. The term "ROS wave" is mentioned several times without explanation. Readers unfamiliar with the specific jargon might find it helpful to have a glossary.
Response: Thanks for the note. Added explanation to the term ROS wave
Although the article discusses many ROS-related processes, the transition between sections could be improved. A sentence describing ROS formation in a single compartment of the cell may also help guide the reader through the narrative. The roles they play in signaling could also be described by these sentences. Furthermore, some sections, like those discussing ROS regulation in mitochondria and chloroplasts, seem repetitive.
Response; Thank you for your valuable suggestions. I agree with the comments. Sorry for some repetitions. The oxidation-reduction processes that occur in chloroplasts and mitochondria are the most important, they have many differences, although there are common processes, so there are repetitions.
The conclusion should emphasize specific gaps in knowledge regarding ROS signaling and plant stress responses. As opposed to briefly mentioning the need for further research.
Response: Thanks for the comments. I have added several sections to the text concerning signaling, one of the signal transmission mechanisms. There are quite a lot of unresolved problems in the process of AFK signaling, so I did not dare to list them.
There are many relevant studies cited in the review, but some are out of date. Incorporating more recent research, particularly from the past five years, would improve the review's accuracy.
Response: Thank you. I have used outdated references because they are fundamental in certain areas of knowledge. However, as per your request, I have added newer references.
Make your paper more appealing to a wider audience by emphasizing the practical implications of the findings. Learn how ROS mediated acclimatization can be used to develop crops that are stress-resistant or improve farming practices under harsh conditions.
Response: Thank you for the suggestion. Although the practical use of this knowledge is difficult at present, breeders can benefit from the knowledge of the two phase waves of ROS. I have made an attempt to suggest this as a practical use.
Due to the length and complexity of the sentences in this article, readers may have difficulty following the main ideas. Condensing and simplifying sentences will enhance reading. It is sometimes difficult to agree on the subject-verb agreement as well as verb tense consistency. Consistently use past and present tenses if the article discusses past research findings rather than general principles.
Response: Thank you for your suggestions on how to improve the English language. After editing the manuscript, it will definitely undergo English editing.

Reviewer 3 Report
Comments and Suggestions for Authors
The comments on the review paper by Larisa I. Fedoreyeva. “ROS as signaling molecules to initiate the process of plant ac-climatization to abiotic stress”
The review is aimed to describe and summarize the processes occurring during ROS homeostasis in plants and the participation of ROS in signaling processes activating adaptation processes to abiotic stresses. ROS are very interesting and complicated topic, as redox reactions are not intuitively understandable and demands higher than basic knowledge in chemistry. It is widely accepted that ROS serve as critical signaling molecules and play a central role in initiating and regulating plant acclimatization to all abiotic stress conditions. There are a lot of reviews about ROS in plants and relation stresses, so it was interesting what is the essence of the new review.
The beginning chapters were interesting to read, but after finishing of reding it was quite difficult to get any clear picture of involving of ROS in abiotic stress acclimation. In general, introduction contain all the essence of the review and following chapters did not add clearance into the topic.
Chapters 2 is fine as summarized generally available information about ROS species. But throughout the description of different ROS species, it is not presented author view about the importance and role of different ROS.
Figures 3-5 here look strange and better to present them as a table with short description and/or biological explanation of the reactions.
Very few info about the most damaging hydroxyl radicals, especially their role in lipid membranes damaging was not mentioned.
Chapter 3 and 4 looks like a collection of citation from several papers, often repeating each other, when different paragraphs repeating the same information with different wording, like it was in the cited papers. The presented information is not structured by the authors understanding of the redox reactions and ROS species in different cellular compartments. So, it is very difficult to follow the topic, especially for the new reader wishing to learn about ROS.
Chapter 5 is also superficial and do not present structural view on the role of ROS under stresses.
Figure 7 looks strange, and it is better to present it as the table with references and more detailed description. Figure 8 looks unfinished and too sketchy; organelles are not indicated. What is SOH is not mentioned anywhere.
The most recent reviews papers related to ROS in plants were not considered, but maybe they could help authors to present their original view on the topic.
I do not think that in current state this review will add anything for the better understanding of the role of ROS in abiotic stresses.
Author Response
Thank you for your careful review of the manuscript and critical comments.
Chapters 2 is fine as summarized generally available information about ROS species. But throughout the description of different ROS species, it is not presented author view about the importance and role of different ROS.
Response: Thank you for your comment. Indeed, ROS is one of the most popular areas of research and many interesting reviews are devoted to it, which are considered from different points of view. The selection of material is already my view and my opinion on this topic. In this area, I am more interested in oxidation-reduction reactions, signal transmission mechanisms. Since all changes can only be identified by comparing with various stress effects, therefore, abiotic stresses were used as a tool. A lot of material has accumulated on each abiotic stress by now, but it is not enough to finally clarify the signal transmission mechanisms for each stress and, most importantly, how a plant determines the type of stress and triggers a specific cascade of interactions.
Figures 3-5 here look strange and better to present them as a table with short description and/or biological explanation of the reactions.
Response: Thanks for the comment. These figures have been reworked and presented as reaction equations
Very few info about the most damaging hydroxyl radicals, especially their role in lipid membranes damaging was not mentioned.
Response: I agree with you. There was little data on hydroxyl ions in the text. Hydroxyl ions are substrates for the hydroxylation of many molecules and can also play a positive role. However, they are generally toxic and have a destructive effect on many molecules. However, there is little data on their participation in the signaling process.
Chapter 3 and 4 looks like a collection of citation from several papers, often repeating each other, when different paragraphs repeating the same information with different wording, like it was in the cited papers. The presented information is not structured by the authors understanding of the redox reactions and ROS species in different cellular compartments. So, it is very difficult to follow the topic, especially for the new reader wishing to learn about ROS.
Response: Thanks for the note. Section 3 provides general information on oxidative stress. This may be why there are repetitions in Section 4.
Chapter 5 is also superficial and do not present structural view on the role of ROS under stresses.
Response: Thank you for your comment. Indeed, this manuscript did not intend to present in detail the action of ROS under abiotic stress. Abiotic stress was used only as a tool to detect the formation of ROS and the ROS signaling wave to initiate the acclimatization process.
Figure 7 looks strange, and it is better to present it as the table with references and more detailed description. Figure 8 looks unfinished and too sketchy; organelles are not indicated. What is SOH is not mentioned anywhere.
Response: Thank you for your comments. Figure 7 has been removed, it really doesn't provide much information in this context. Figure 8 has been revised. In addition, several sections have been added to the text, one of which contains information about the formation of the SOH group in proteins.
The most recent reviews papers related to ROS in plants were not considered, but maybe they could help authors to present their original view on the topic.
Response: Thank you very much for the suggestion. Several sections have been added to the manuscript, which list contemporary review publications.
